# Enablers and barriers to effective HIV self-testing in the private sector among sexually active youths in Nigeria: A qualitative study using journey map methodology

**Dennis Aizobu[1], Yusuf H. Wada[1]\*, Jennifer Anyanti[1], Godpower Omoregie[1], Boluwatife Adesina[1], Serah Malaba[2], Morghan Kabeer[3], Samuel Oyegunle[3], Akudo Ikpeazu[4], Omokhudu Idogho[1]**

**1** Society for Family Health, Abuja, Nigeria, **2** Population Services International, Nairobi, Kenya, **3** Busara Center for Behavioral Economics, Nigeria, **4** National AIDS/STIs Control Programme, Abuja, Nigeria

\* hwada@sfhnigeria.org

**Data Availability Statement:** Data cannot be shared publicly because of the nature of the study and national ethical restrictions on data which

## Abstract

### Background

HIV is a public health burden in Nigeria. HIV self-testing is one of the approaches to testing, which is the first of the 95:95:95 cascade of a coherent response to the epidemic. The ability to self-test HIV is influenced by various factors that can either serve as enablers or barriers. Exploring these enablers and barriers to the uptake of HIVST will help achieve optimal HIV self-testing and provide a deeper understanding of the HIVST kits users' journey.

### Objective

The purpose of the study was to identify enablers and barriers to the uptake of HIV self-testing among sexually active youth in Nigeria using journey map methodology.

### Methods

We conducted a qualitative exploratory study between January 2021 to October 2021 to understand the journey map for taking up and using HIVST in the private health delivery systems which include the pharmacies and PPMVs. 80 youths in Lagos, Anambra and Kano states were interviewed using IDIs and in-person FGDs. Their responses were audio-recorded, transcribed and analyzed using a qualitative software package (Nvivo software).

### Results

A journey map for taking up and effectively using HIVST using the private sector among sexually active youths using key enablers and barriers at the attract, purchase, use, confirmation, linkage, and reporting stage was developed. The major enablers among participants were privacy and confidentiality, bundling purchases with other health products, easy-to-use instructions, and past experience with other self-testing kits. The major barriers were

includes potential identifiable voice and names through the FGDs and IDIs and contain/identify potential sensitive information of the stakeholders, identify HIVST products which have influence on the participants safety. They were also assured that data will only be anonymously analyzed and also shared only on reasonably request as approved on the research protocol. Data are available from Society for Family Health Institutional Data Access for researchers who meet the criteria for access to confidential information through a non-author point of contact email info@sfhnigeria.org or through the corresponding author email hwada@sfhnigeria.org."

**Funding:** This study was possible by the generous support of Children's Investment Fund Foundation (CIFF) - Global Fund partnership through the Population Services International (PSI). The contents herein are the sole responsibility of the authors and do not necessarily reflect the views of SFH, PSI or CIFF. The funders had no role in study design, data collection and analysis, decision to publish or preparation of the manuscript.

**Competing interests:** The authors have declared that no conflict of interests exists.

fear of discrimination, big packaging, high price, lack of confidence from user error and fear of status disclosure.

## Conclusions

**Sexually** active young people's perspectives enhance our understanding of the barriers and enablers of using HIVST through the private sector. Optimizing the enablers such as improved confidentiality that may be seen in e-pharmacy, reducing barriers and factoring sexually young people's perspectives will enhance the market and the uptake of HIVST towards ensuring sustainability and accelerating progress towards the 95-95-95 targets.

## Background

HIV is a public health burden in Nigeria and worldwide. It remains devastating for adolescents and young adults aged 15–24 years, who bear a disproportionate burden of approximately 50% of all new HIV infections and 33% of persons living with HIV/AIDS globally [1]. In Nigeria, the prevalence of HIV is 1.4% among the general adult population, with about 1.9 million people living with HIV/AIDS as of 2018 [2]. Because of the tremendous threat HIV poses to public health, there remains a high proportion of key populations, especially sexually active young people, unaware of their HIV status [3]. These young people are at risk of exposing their sexual partners to new HIV infections, which threaten to undermine progress made in addressing the HIV epidemic. In contrast, studies have shown that the desire to be tested for HIV among adolescents in Africa is relatively high, with approximately two-thirds of untested adolescents desiring an HIV test [4].

Due to a lack of awareness of the importance of HIV testing, there is low initiation of antiretroviral therapy (ART), which naturally hinders the success of global initiatives intending to eliminate HIV by 2030 [5]. Therefore, broadening screening opportunities by increasing access to HIV self-testing (HIVST) as part of a comprehensive method mix is a key strategy for reducing the prevalence and threat unknown HIV status poses in low-income countries. To help complement decreasing trends in HIV incidence, various countries, including Nigeria, have recently rolled out HIVST, especially those that use blood or saliva (oral HIV test) to test for HIV. The private sector delivers HIV prevention and care services to communities and serves as primary health delivery systems [3]. This includes the community pharmacies (CPs) and patent and proprietary medicine vendors (PPMVs) who provides HIV prevention services such as the HIVST kits and sexual and reproductive health (SRH) products and services such as condoms, lubricants and antiretrovirals [4]. However, due to low confidence from users in using HIVST, poor linkage to care, and low-risk perception in administering the HIVST, some young populations do not trust the accuracy of HIVST results [6]. These highlights the importance of documenting the barriers and enablers faced by young persons in the uptake of HIVST in Nigeria, especially with the involvement of private sector distribution to allow for self-care.

However, despite the large-scale programmatic approach to increase the prevalence of self-testing in Nigeria, multi-level barriers to the use of HIV testing among Nigerian youths have been reported at the individual (low perception of risk and fear), structural (lack of testing sites, stigma) and social (social support, service-related and limited peer support) level [5–10]. The major barrier to HIV testing and counselling cited by untested adolescents and young adults was fear, including parents' reaction to a positive result, fear of needles, stigma/

discrimination, AIDS-related illness and death [6, 11]. The involvement of private sector distribution of HIVST to allow for self-care requires key documentation of specific barriers and facilitators which could drive test demand, the usability of tests, and managing pricing to ensure broader access, mitigate transmission, and increase HIVST implementation through policy changes, programmatic approach, advocacy and research.

Furthermore, the research evidence regarding factors that enable and affect the uptake of HIVST, especially among youths globally and Nigeria in particular, remains limited. Only few studies have been able to explore the young people's preferences for HIVST service delivery in Nigeria [7] and in sub-Saharan Africa [12–14]. Therefore, in response to the above seemingly grim picture and to expand access to HIVST within the framework of informed choice, Society for Family Health (SFH) and partners: Population Services International (PSI) and National AIDS and STIs Control Program (NASCP), aimed to gather insights and identify gaps in HIV self-testing in Nigeria, and explore other factors which serve as enablers and barriers for the uptake of HIVST kits through the private sector. These will help to achieve optimal HIV self-testing and determine the journey of HIVST kits for users, providing a deeper understanding of the key moments along the journey: from attracting stage, purchase, use of test, confirmatory and linkage to prevention, care and treatment, where problems may occur.

## Methods

### Design

In this study, a descriptive qualitative study was used to explore insights on enablers and barriers to the uptake of HIVST kits. PSI and SFH conducted qualitative market research on HIV self-testing (HIVST) in the private health delivery systems (pharmacies and PPMVs) in Nigeria. The study was conducted in 3 PEPFAR priority states of Anambra, Lagos and Kano and represents the key geographical mapping in terms of HIV prevalence, unmet needs for HIV/AIDS treatment services and actual size estimation. This helps to understand the environment for users, how they interact with the private health delivery systems and the enablers and barriers to intervention on both the demand and supply aspects.

### Journey map operation

The participant journey-map was adopted from studies [15–17] and used to explore the pathway to identify potential pivot points, link encounters, understand perspectives that facilitate insights, promote proactive delivery of people-centered care, and improve study reliability. Our method, therefore, explored this approach to gain insights into the participants' processes before taking up and using HIVST in private sector. This was subsequently used to create a narrative of the sexually active users' thoughts and emotions, leading to visualization of their actions when taking up and using the HIVST kits.

In this study, we distilled the journey for taking up and using HIVST kits through the attract, purchase, and use stages and later diverges based on their perceived HIVST results. For those who receive a non-reactive result, they proceed to link through the HIV prevention continuum [18–20]. They begin with a link to the HIV prevention services stage followed by retention in such services and later adherence to prevention interventions with repeated testing for HIV acquisition monitoring. While those who are perceived to have a reactive result proceed to the confirmatory testing, link to HIV care and treatment, and reporting stage. At each step, many factors inhibit their willingness and ability to use HIVST, which was reported.

## Recruitment and participants selection

The study involved sexually active youths of different genders, residences, HIV status and past experience of using HIVST across Nigeria for FGDs and IDIs. We conducted 45 IDIs and 7 FGDs with 80 sexually active males and females across Anambra, Kano and Lagos states. IDIs and FGDs respondents were reassured that data that could potentially identify a person or location was anonymized and could not be traced back to them. Electronic data were password protected.

The recruitment of participants was carried out using clearly spelt-out eligibility criteria and a convenience sample drawn at random from a list of participants provided by community-based organizations (CBOs) and civil society organizations (CSOs) who fulfilled the criteria and were willing to write informed consent.

The inclusion criteria were: 1) sexually active young people between 18 and 29 years residing in urban or peri-urban settings in Nigeria that have either taken an HIVST or not. 2) willing and able to provide verbal informed consent and consent to an audio-recorded session.

## Data collection

Data was collected through validated IDIs (S1 Appendix) and FGDs guides (S2 Appendix) that was developed after an extensive search of published literature [12, 14] and translated into the four dominant languages of Nigeria (English, Yoruba, Igbo, and Hausa). They were then validated through face and content validity to encourage the natural flow of conversation. The face validity was assessed by a team of experts from the Busara Centre for Behavioral Economics, SFH, PSI, Department of Public Health, Federal Ministry of Health, NACA and NASCP with relevant experience in the field of HIV. The content validity was also assessed by an independent team of experts (public health, epidemiologist, infectious diseases physician, and languages expert from the different languages used). The independent team of experts suggested some modifications including the flow and paraphrasing some sentences to be clear, unambiguous, essential and relevant. Thus, this gave us a final copy of our guide used in this research.

The project team from SFH and Busara who were part of the face validity team carried out the qualitative research and collected data in the four dominant language of Nigeria (English, Igbo, Hausa, and Yoruba) using the IDIs and FGDs guides. IDIs were done in English over the phone, while FGDs were conducted in-person for the sexually active youths using the four languages in the three states. We conducted gender-specific FGDs—one FGD was either all men or all women, but we mixed HIVST users and HIVST non-users. During both the IDIs and FGDs, we asked respondents open-ended questions regarding their opinions and perceptions of HIVST, and for IDIs with HIVST users, we asked questions about their past experiences with taking up and using HIVST kits. Triangulation of data from three sources (focus group transcripts, past experience detailing enablers and barriers, and individual questionnaires of sociodemographic) and interviews complemented involving individual prioritization of the barriers and facilitators to HIVST [21, 22].

Using information collected, we identified enablers and barriers to HIVST kit uptake and use, and then systematically mapped these enablers and barriers onto the journey map for taking up and using HIVST kits in the private sector, allowing us to clearly identify opportunities to leverage and challenges to overcome.

## Data management and analysis

All data analysis were done based on Graneheium and Ludman system to assure trustworthiness [17]. Interviews were audio-recorded and transcribed verbatim. In-depth interviews (IDIs) and focus group discussions (FGDs) were audio-recorded, transcribed, and translated

into English. Thematic analysis (TA) was used to analyse and code the data using a qualitative software package—Nvivo. Using information from the IDIs and FGDs, we identified enablers and barriers to taking up and using HIVST.

### Ethical considerations

This study was approved by the two ethical review committees, the National Health Research Ethics Committee of Nigeria (NHREC)–NHREC/01/01/2007-29/06/2021 and the Nigerian Institute of Medical Research (NIMR)–IRB/21/043. Permission was also sought from the study participants through written informed consent. This was done after the benefits and objectives of the study had been fully read by them and all clarifications provided. All research activities were conducted in accordance with the Declarations of Helsinki and other policies and regulations required by the ethical committees. All respondents consented to participate, and they were free to voluntarily withdraw at any time with no attraction of penalty at any time during the research.

## Results

In total, 80 participants were enrolled in the study, with majority being male (55%), at the age range of 21–25 (43.8), single (92.4), educated at tertiary level (57.4) and living in urban areas (52.5) [See **Table 1**]. FGDs were of the same gender, but groups were a mix of peri-urban/urban residents, HIV status, and past experience with HIVST. Majority of the IDIs were from sexually active youths who reside in Lagos state (17) with 3 FGDs that had a good representation from the urban and peri-urban settings. Majority of these IDIs have used HIVST in the IDIs, while there was a mix of both users and non-users for the FGDs. Anambra and Kano states have the same IDIs (14 each) and FGDs (2 each) representing males and females, both urban and peri-urban residents, people living with HIV and HIV-negative persons and users of HIVST and non-users [**Fig 1**].

### Enablers and barriers to taking up and using HIVST in the private sector

Description of the sexually active young people journey map:

In this study, seven individual steps were mapped out in detail and then condensed into a single journey map. The steps are the attract, purchase, use, confirmation testing, link to HIV care and treatment, link to HIV prevention and reporting stage, which are all crucial in developing effective and sustainable HIV self-testing care.

Throughout the entire journey map:

Fig 2 shows the enablers and barriers for taking up and using HIVST Kits in the private sector at various stages. Most sexually active males and females from our participants were aware of the factors that put them at risk of HIV, and most possessed the self-awareness to know that their behavior may put them at risk. Key themes as stages were highlighted, which shows the major enablers and barriers at every important stage which might either undermine or enable the use of HIVST in the private sector.

### Enablers and barriers at the attract stage

The major enablers that attracted sexually active youths to take up and use HIVST were: influence from friends in the health sector, online/social media or print and radio advertisements, awareness of the need for HIV testing, privacy/confidentiality of testing at home, community sensitization with friends and curiosity on alternative ways of testing. While the barriers included fear of

**Table 1. Demographic characteristics of participants.**

| Demographic variable | Category | Frequency (n) | Percentage (%) |
|---|---|---|---|
| Age | 18–20 | 19 | 23.75 |
| | 21–25 | 35 | 43.75 |
| | 26–29 | 26 | 32.50 |
| | Total | 80 | 100 |
| Gender | Male | 44 | 55.00 |
| | Female | 36 | 45.00 |
| | Total | 80 | 100 |
| Marital Status | Single | 74 | 92.50 |
| | Married | 5 | 6.25 |
| | Divorced | 1 | 1.25 |
| | Total | 80 | 100 |
| Highest educational level | Primary education | 5 | 6.25 |
| | Secondary education | 29 | 36.25 |
| | Tertiary education | 46 | 57.50 |
| | Total | 80 | 100 |
| Geographic distribution | Urban | 42 | 52.50 |
| | Peri urban | 38 | 47.50 |
| | Total | 100 | 100 |
| Interviews | FGDs | 45 | 86.54 |
| | IDIs | 7 | 13.46 |
| | Total | 52 | 100 |

discrimination from buying kit in public, perceived lack of awareness among peers, fear of reactive result, low-risk perception among peers and avoiding being treated as sick.

## Enabler: Privacy/confidentiality of testing at home

The key enabler, which almost all participant related to, was the privacy and confidentiality of testing at home. Participants were attracted to the idea of testing at home and in private, without any influence from external parties. So, no one would know they took the test, and they could keep their results to themselves.

> *"What really got my attention from the onset was the fact that you can do it at home without anybody knowing about your status, so it was easy to make the decision to take [the HIVST kit]."*
>
> *- Male, Lagos, 25yrs, Peri-urban*

## Barrier: Fear of discrimination from buying kit in public

The majority of participants at the attracting stage were afraid of being judged for buying the kits at a pharmacy by people within the community. They said they either would not buy the kit at a pharmacy or would go to a pharmacy in a neighboring community.

**Figure 1:** The demographic characteristic of FGDs and IDIs with Sexually Active Males and Females in Anambra, Lagos and Kano states of Nigeria

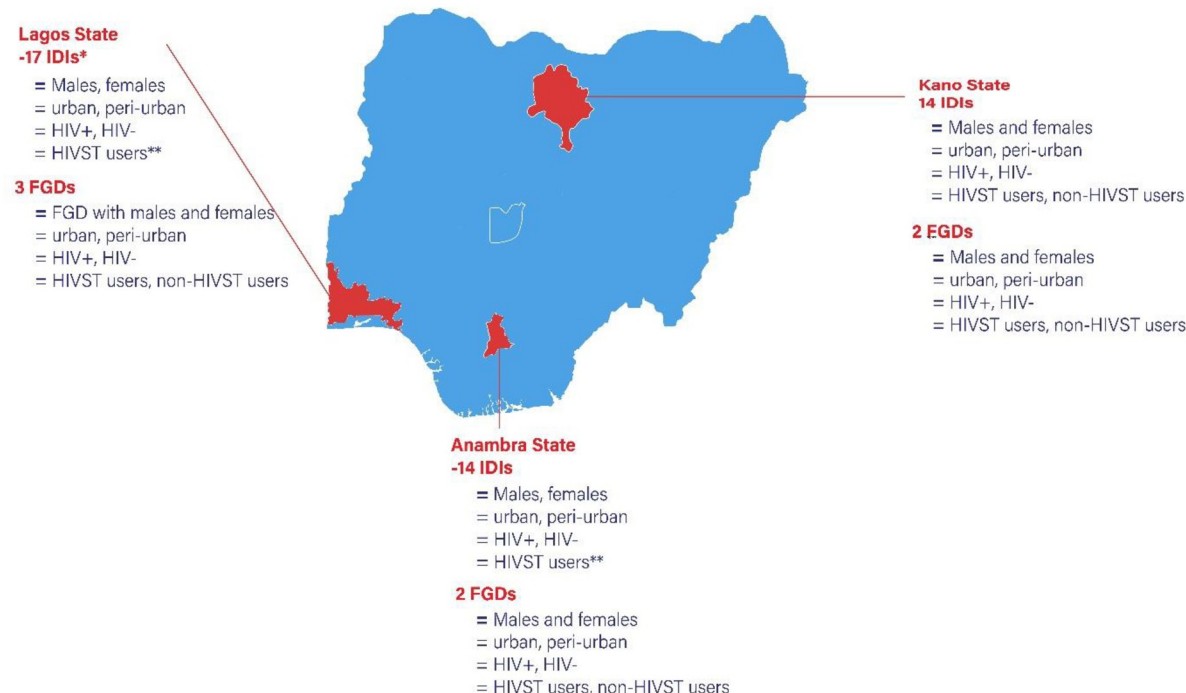

**Fig 1. showing the demographic characteristic of FGDs and IDIs with sexually active males and females in Anambra, Lagos and Kano states of Nigeria.**

*"That [pharmacy] won't be comfortable for me considering how most people in the society view PLWHIV as sexually promiscuous individuals. Also, there are occasions where I get to meet attendants who are not professional in keeping customers' status confidential. . .."*

*Male, Anambra, 25yrs, Urban*

## Enablers and barriers at the purchase stage

The key enablers identified by the participants were purchase and use at customers' convenience, bundling purchasing with other health purchases, privacy with face masks and COVID-19, habitually buying condoms at pharmacies, online pharmacies in urban areas, and limited hassle factors to obtain HIV testing services. Contrastingly, the barriers in the purchase stage of HIVST in the private sector highlighted by the majority of participants include the convenience of subsequent care at health center, lack of awareness that kits are sold at pharmacies, insincere and untrustworthy pharmacists, big packaging and high price.

## Enabler: Bundling purchase with other health products

Participants said buying HIVST at a pharmacy would be convenient, as they usually buy other health products at pharmacies, which they can also purchase while buying HIVST.

Figure 2: *Enablers and Barriers for Taking Up and Using HIVST Kits in the Private Sector*

**Fig 2. Enablers and barriers to taking up and using HIVST kits in the private sector.**

*"Yes, it is very convenient to buy [HIVST] from the pharmacy mainly because the pharmacy is a place where you can get medical products for general public consumption. Hospitals may not be easily accessible."*

*- Male, Anambra, 24yrs, Urban*

## Barrier: Price

Participants thought the kits were too expensive. They were willing to pay between 100–1,000 Naira (US$ 0.2–2) for the kit.

*"Myself and a lot of others also complained of the cost of the kit and considered it quite expensive. We felt most young people may not be able to afford it."*

*- Male, Anambra, 28yrs, Peri-urban*

### Enablers and barriers at the use stage

Overall, the majority of the participants thought HIVST kits were easy to use. They reported major enablers of using HIVST as having easy and salient instructions, past experience with other self-test kits, support from close friends, self-confidence and self-efficacy, painlessness of oral-fluid-based tests, autonomy, and confidence in HIVST efficacy. The major barriers were lack of confidence in HIVST from user error, lack of counseling from healthcare professionals, lack of confidence in HIVST from unexpected results, skepticism of oral-fluid-based kits and perception of doctor's abilities.

### Enabler: Autonomy

Majorly all participants liked that they could use the test where and when they wanted.

> "Seeing the kit gives me joy. At least I can do it on my own, do it for anybody. I can just do it at my house, even in my office. I can do it anywhere."
>
> - Female, Kano, 29yrs, Peri-urban

### Barrier: Lack of counseling from healthcare professional

Most participants lamented that when doctors administer an HIV test at a health facility, they comfort and counsel patients upon receiving their results. Self-testing in private does not offer these counseling services that are often inherent to testing at a health facility.

> "It shows that it's HIV, but what I heard from the doctors is that it is not the worst...when I tested and it was confirmed that there is HIV, then I said AIDS, and he said no, it is HIV, and he said when I keep taking care of myself, I will be fine and ok."
>
> - Female, Kano, 28yrs, Peri-urban

### Enablers and barriers at the confirmatory testing stage

Participants knew they were supposed to go to the health facility for confirmatory testing should they receive a reactive HIVST result. The majority of them said that the major enablers for them at the confirmatory testing were that the HIVST had clear instructional materials, support from close friends or siblings, their perceptions of doctor's abilities, seriousness of reactive results and perceived responsibilities of doctors. They also highlighted key barriers at the confirmatory testing stage that included a lack of salient instructions on the next steps, lack of follow-up from healthcare providers, delays caused by shock and denial, comfort with non-reactive results, and lack of money for transportation.

### Enabler: Seriousness of reactive results

Participants who tested reactive trusted that the results were accurate and took the results seriously, and as a result, went for confirmatory testing. Several participants implied that it was clear that they needed to go for confirmatory testing upon receiving a reactive result.

*"Of course, I went for confirmation. That is when they asked me to come back in the next 3 months."*

*- Male, Kano, 29yrs, Urban*

### Barrier: Delays caused by shock and denial

Some participants were in shock and denial over their HIVST results and waited up to two months to go for confirmatory testing.

*"I felt ashamed of the result, so I didn't want to go [for confirmatory testing]. I felt embarrassed. I felt that something could be changed through prayers."*

*- Female, Anambra, 21yrs, Peri-urban*

### Enablers and barriers at the link to HIV care and treatment stage

Many participants were able to seamlessly link to HIV care and treatment immediately after receiving the results from their confirmatory test. But they also reported that experiencing poor health, seamless linkage to care after confirmatory testing, and knowledgeable healthcare centers were the main enablers of HIVST in the private sector. The major barriers limiting the linkages to HIV care and treatment after using HIVST were delays caused by stress and embarrassment, perception of unfriendly healthcare staff, avoidance of being treated as sick and fear of status definition.

### Enabler: Seamless linkage to care after confirmatory testing

After participants went to the hospital for confirmatory testing, they were started on treatment immediately and didn't need to go back to the health center for another appointment. This process minimized drop-off.

*"I went to a health facility [after using HIVST] and got a confirmation test and treatment."*

*- Male, Lagos, 21yrs, Urban*

### Barrier: Delays caused by stress and embarrassment

Positive results after confirmatory testing evoked strong emotions of stress and embarrassment, leading some to delay seeking care.

*"I started since when I was told that I am [positive]. I cried and went home. It was [my friend], she took me to one hospital, we went there, and we were attended to. [They] welcomed me in a nice way. I was interviewed by a doctor, and he gave me medicine."*

*-Female, Kano, 28yrs, Peri-urban*

### Enablers and barriers at the link to HIV prevention stage

After receiving a non-reactive HIVST result, the vast majority of participants did not do anything. To most, a non-reactive result was validation that their HIV prevention methods were effective, and they thought their knowledge of HIV prevention was sufficient.

### Enabler: Awareness of need for HIV prevention

Participants were aware of the adverse effects of the disease and were self-aware enough to know they may be at risk. As a result, they consciously sought information to protect themselves, either by abstaining or using condoms. Many of these participants have friends that work in the health sector and/or have attended sensitizations on HIV.

> *"I usually seek information. I usually browse on my phone, read books [about] prevention and all that, and I read that one of the preventive measures is condom. Once you are using condoms, although you are not fully assured because sometimes it can burst, but I believe it is the number 1 preventive measure."*

> *- Female, Anambra, 25yrs, Peri-urban*

### Barrier: Perceived knowledge of HIV prevention strategies

After using HIVST and testing non-reactive, participants did not seek out HIV prevention strategies. Most participants use condoms, and some use PREP. Most assumed their strategies were effective or they already knew everything they needed to know about HIV prevention.

> *"No [I didn't seek information on HIV prevention] because I am an adult, and I already know about HIV prevention."*

> *- Female, Lagos, 25yrs, Urban*

### Enablers and barriers at the reporting stage

Many participants acknowledge the importance of reporting their HIVST results. Some said they would consider reporting their results to facilitate linkage to HIV prevention or treatment and care:

> *"I think potential benefits [of reporting HIVST results] is getting access to the prevention services like PrEP."*

Although many participants recognized the importance of reporting, many said they would not report their HIVST results to anyone, largely due to issues with privacy and confidentiality. They did not want many people to know their HIV status.

As one respondent stated: *"If my result was positive, I think the negative consequences [of reporting my results] could be losing confidentiality over my health status."*

### Enabler: Easy to link to HIV treatment/prevention

Participants thought the main benefit of reporting their results more widely to other entities besides their hospital/health center would be access to better prevention measures or more immediate access to treatment.

> *"Yes, because I reported a negative result, I think potential benefits getting access to the prevention services like PrEP."*
>
> *- Male, Lagos, 28yrs, Peri-urban*

### Barrier: Concerns over data privacy

Participants were concerned about data privacy; if NGOs, healthcare facilities, and other organizations were careless about data privacy, someone could see their results.

> *". . .if you carelessly keep it and somebody comes and finds it."*
>
> *- Male, Kano, 29yrs, Urban*

### Insights into the enablers and barriers for non-users of HIVST

Our findings reported that the majority of participants who were classified as non-users said that they did not think they needed HIVST, nor were they aware of where they could acquire HIVST. They also reported that they might want support from a close friend or sibling to use HIVST.

They also reported a low-risk perception at the attracting stage, with non-users saying they have never used HIVST because it never occurred to them or they never thought they needed it.

At the purchase stage, the majority of the participants reported a lack of awareness that kits are sold at pharmacies which might hinder the use of the HIVST. These non-users were unaware that they could get HIVST at a pharmacy, but they would be open to buying the kits for 200–500 Naira. Similarly, the majority highlighted self-efficacy and self-confidence at the use stage. Non-users have experience with self-testing, namely pregnancy self-test kits, and they felt confident with following the instructions and administering the test. At the confirmatory stage, there was a strong perception of support from close friends or siblings. Majority stated that *"if they tested reactive, they would first tell a close friend or family member"*, then go for confirmatory testing. Perceived knowledge of HIV prevention strategies and seamless linkage to care after confirmatory testing were reported at the linkage to HIV prevention stage and HIV care and treatment stage, respectively. At the reporting stage, the majority reported a lack of knowledge on the next steps for non-reactive results.

## Discussion

To the best of our knowledge, this is the first study that has reported on the enablers and barriers to the uptake and use of HIVST in Nigeria with a good representation of the key geographical regions using a journey map. This is also the first study that reported a comprehensive report on the journey map for taking up and using HIVST in the private sector among sexually active males and females in any country of the world. Furthermore, this study is also the first

on HIVST among youths that segregated the barriers and enablers based on every stage (attracting stage, purchase, use, confirmatory testing, linkages to care, prevention, treatment and reporting) that are key to generate demand.

There are roughly 500,000 PLHIV in Nigeria who do not know their HIV status [1]. HIVST offers an innovative new way to provide high-risk populations with HIV testing services. Only a few pilot experiments on HIVST have been implemented in West African countries, such as our sample cohort, Nigeria [23]. Through our IDIs and FGDs with sexually active youths, we uncovered many in-depth insights on taking up and using HIVST in the private sector. At a high level, we found that youths see value in having access to reliable HIV testing services, such as HIVST, as they are well-informed about HIV and its risk factors, which they learned from HIV sensitizations organized by non-governmental organizations (NGOs) and health workers in their communities. Likewise, young people like the convenience, accessibility, discretion, autonomy and ease of using HIVST kits whenever and wherever they please. This is consistent with studies conducted among young populations in many sub-Saharan African countries [7, 12, 23–25].

Our study finds out that young people showed high acceptability and preferred the private-driven model over the public-driven model in the provision of HIVST kits. The reasons attributed to these were due to high accuracy and access to high-quality of HIVST in the private facilities. This is consistent with a study which reported young people's preference for HIVST in Nigeria [7, 26]. However, their study only reported young people preference to the private sector whereas ours reported on both preference and usability. Young people at each stage reported fear of discrimination, perceived lack of awareness, high pricing, status disclosure, linkages to counseling services as key concerns. For policymakers and regulators, it is essential that young people are interested in using HIVST. Confidentiality, out of pocket payment and information plays significant roles in their decision to buy HIVST kits. While this clearly shows a market for HIVST, the drivers are limited. There is a need to eliminate these social and psychological barriers and explore the opportunities that exist [14].

An interesting result from our study was that many young people preferred home-delivery, low-cost and value HIVST for providing greater discretion around their sexual debut at the attract stage. This is consistent with a study of young people in Malawi and Zambia [12]. For example, at the attract and purchase stage, there was also a high perceived risk of fear of discrimination from buying a kit in public and fear of reactive results or being judged as a sick person. This is an important issue that needs to be addressed, especially leveraging the use of online pharmacy services approved recently by the Pharmacists Council of Nigeria, which can drive the provision of home delivery self-testing kits at one's convenience. Additionally, this has important implications for policy design and full optimization of not only HIVST opportunities but also the entire self-care policy drive in Nigeria.

Further, the scalable strategies for confirmation of their HIV status and linkages of HIV prevention, care and treatment need to be optimized and so much needed to maximize individual and public health benefits of HIVST [27]. Confidence in healthcare professionals also encourages young people to seek confirmatory testing after using HIVST, as youths trust healthcare professionals to properly administer a confirmatory HIV test and provide them with adequate HIV treatment and care. However, high perceived trust in health care professionals, such as doctors, also posed a barrier to HIVST uptake and use in the private sector. Similarly, HIV testing at the health facility is free, and many youths think the price of HIVST at the pharmacy is unaffordable. Besides being too expensive, youths are reluctant to purchase HIVST at a pharmacy or PPMV because they don't trust the pharmacist or PPMV owner to keep their purchase confidential, and they would not want other customers to see what they're buying. This implies that young people are willing to pay if they believe the prices, they pay

signal more quality of care and services, such as the provision of accurate information, and if they trust their doctors. These findings are in line with many reported studies which shows that willingness to purchase and low ability to pay for HIVST kits are associated with quality of care and unique value effect of the product across sub-Saharan Africa [12, 28–35].

Therefore, evidence of new delivery models, such as social business marketing on affordable and effective HIVST delivery models for increasing testing coverage among key populations such as young people, underserved populations and high-risk populations, are necessary for country decision-making in countries like Nigeria [21]. The social business enterprise model through a total market approach has been used for many reproductive health products, such as condoms, to expand access to high-quality and low-cost products to fulfil important needs of the underserved population. Additionally, the majority of HIVST users and non-users reported that self-efficacy and self-confidence at the use stage were key enablers of HIVST. This is consistent for HIVST and other self-testing kits, such as pregnancy self-test kits, which they felt confident with following the instructions and administering the test. It would also be important to expand this instruction to other dominants language such as Hausa, Igbo and Yoruba for a wider audience coverage.

## Limitation

Our study was a qualitative, multi-centered study with the largest sample size among similar studies done in Africa with similar objectives. However, it does have limitations that might be subject to social desirability bias. Our methodological approach led to a few limitations that affected the interpretation and application of our results. We used convenience sampling to recruit sexually active participants, which may introduce bias. However, we took steps to minimize sampling bias by the use of multiple sites, using indirect questions, random sampling from pool of participants presented, inclusion of both the users and non-users and ensuring the sufficient sample size to enable the applicability of results. Many sexually active males and females were recruited from HIV prevention, care, and treatment programs and other existing HIVST programs. As such, they were aware of HIV risk factors, testing methods, and prevention strategies, and as a result, more likely to recognize the need for HIV testing. Their knowledge and perceptions are reflected in the answers they provided during the IDIs and FGDs and may not represent the average sexually active young person in Nigeria, who is not currently involved in any HIV or HIVST program.

## Conclusion

This paper offers an overview of the journey map for taking up and using HIVST in the private sector. Optimizing the enablers, such as improved confidentiality, that may be seen in e-pharmacy, reducing barriers and factoring sexually young people's perspectives will enhance the market and the uptake of HIVST towards ensuring sustainability and accelerating progress towards the 95-95-95 targets. This also impacts health-seeking behaviors, which for HIV has significant ramifications on outcomes on prevention, treatment and viral suppression due to the rapid progression of the disease. Furthermore, an important next step for research will be to explore the possibilities of using an online pharmacy model to address some of these barriers, and the possible economic cost analysis of HIVST distribution modalities, such as social marketing models and secondary distribution, among others. This would enable efficient programs to be formulated to provide linkages with the supply chain, understanding the true picture, cause and stakeholders for key policy actions, advocacy and grow the market through sustainable financing.

## Supporting information

**S1 Appendix. IDI guide, sexually active males and females.**
(DOCX)

**S2 Appendix. FGD guide, sexually active males and females.**
(DOCX)

## Author Contributions

**Conceptualization:** Dennis Aizobu, Jennifer Anyanti, Godpower Omoregie, Boluwatife Adesina, Serah Malaba.

**Data curation:** Dennis Aizobu, Godpower Omoregie, Boluwatife Adesina.

**Formal analysis:** Morghan Kabeer, Samuel Oyegunle.

**Funding acquisition:** Dennis Aizobu, Jennifer Anyanti, Omokhudu Idogho.

**Investigation:** Samuel Oyegunle.

**Methodology:** Boluwatife Adesina.

**Project administration:** Dennis Aizobu, Jennifer Anyanti, Godpower Omoregie, Boluwatife Adesina, Akudo Ikpeazu.

**Resources:** Yusuf H. Wada, Serah Malaba, Morghan Kabeer.

**Software:** Serah Malaba, Morghan Kabeer.

**Supervision:** Dennis Aizobu, Godpower Omoregie, Boluwatife Adesina, Serah Malaba, Akudo Ikpeazu, Omokhudu Idogho.

**Validation:** Yusuf H. Wada, Jennifer Anyanti, Boluwatife Adesina, Serah Malaba, Morghan Kabeer, Akudo Ikpeazu.

**Visualization:** Yusuf H. Wada, Morghan Kabeer, Samuel Oyegunle.

**Writing – original draft:** Yusuf H. Wada, Boluwatife Adesina, Serah Malaba, Morghan Kabeer, Samuel Oyegunle.

**Writing – review & editing:** Dennis Aizobu, Yusuf H. Wada, Jennifer Anyanti, Godpower Omoregie, Serah Malaba, Morghan Kabeer, Akudo Ikpeazu, Omokhudu Idogho.

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
