## [Decision Letter · Decision Letter 0]

3 Nov 2022

PONE-D-22-23769Enablers and Barriers to Effective HIV Self-testing in the Private Sector among Sexually active youths in Nigeria: A Qualitative Study using Journey Map methodologyPLOS ONE

Dear Dr. Hassan Wada,

Thank you for submitting your manuscript to PLOS ONE. After careful consideration, we feel that it has merit but does not fully meet PLOS ONE’s publication criteria as it currently stands. Therefore, we invite you to submit a revised version of the manuscript that addresses the points raised during the review process.

Please respond carefully to all of the points the reviewers have raised when preparing your revision.

A rebuttal letter that responds to each point raised by the academic editor and reviewer(s). You should upload this letter as a separate file labeled 'Response to Reviewers'.A marked-up copy of your manuscript that highlights changes made to the original version. You should upload this as a separate file labeled 'Revised Manuscript with Track Changes'.An unmarked version of your revised paper without tracked changes. You should upload this as a separate file labeled 'Manuscript'

We look forward to receiving your revised manuscript.

Kind regards,

Jamie Males

Editorial Office

PLOS ONE

Journal Requirements:

“This study was possible by the generous support of Children’s Investment Fund Foundation (CIFF) - Global Fund partnership through the Population Services International (PSI). The contents herein are the sole responsibility of the authors and do not necessarily reflect the views of SFH, PSI or CIFF. The funders had no role in study design, data collection and analysis, decision to publish or preparation of the manuscript. We appreciate all members of the SFH SHIPS team, NACA, and ANAYD for their support during the data collection. We are most grateful to all the participants who took part in the study.”

“This study was possible by the generous support of Children’s Investment Fund Foundation (CIFF) - Global Fund partnership through the Population Services International (PSI). The contents herein are the sole responsibility of the authors and do not necessarily reflect the views of SFH, PSI or CIFF. The funders had no role in study design, data collection and analysis, decision to publish or preparation of the manuscript. “

“None”

5. Please ensure that you refer to Figure 1 in your text as, if accepted, production will need this reference to link the reader to the figure.

6. We note that Figure 1 in your submission contain [map/satellite] images which may be copyrighted. All PLOS content is published under the Creative Commons Attribution License (CC BY 4.0), which means that the manuscript, images, and Supporting Information files will be freely available online, and any third party is permitted to access, download, copy, distribute, and use these materials in any way, even commercially, with proper attribution. For these reasons, we cannot publish previously copyrighted maps or satellite images created using proprietary data, such as Google software (Google Maps, Street View, and Earth). For more information, see our copyright guidelines: http://journals.plos.org/plosone/s/licenses-and-copyright.

    a. You may seek permission from the original copyright holder of Figure(s) [#] to publish the content specifically under the CC BY 4.0 license. 

Reviewers' comments:

Reviewer's Responses to Questions

**Comments to the Author**

1. Is the manuscript technically sound, and do the data support the conclusions?

Reviewer #1: Yes

Reviewer #2: Partly

2. Has the statistical analysis been performed appropriately and rigorously? 

Reviewer #1: N/A

Reviewer #2: Yes

3. Have the authors made all data underlying the findings in their manuscript fully available?

Reviewer #1: No

Reviewer #2: Yes

4. Is the manuscript presented in an intelligible fashion and written in standard English?

Reviewer #1: No

Reviewer #2: No

5. Review Comments to the Author

Reviewer #1: “Enablers and Barriers to Effective HIV Self-testing in the Private Sector among Sexually active youths in Nigeria: A Qualitative Study using Journey Map methodology” presents qualitative data from in-depth interviews and focus group discussions with 80 participants involved in market research in Lagos, Anambra, and Kano states from January 2021 to October 2021. Although this is an important area of investigation, the current writing is weak and has numerous grammatical errors, making portions of the manuscript difficult to comprehend. I strongly recommend utilizing the services of a copyeditor to enhance clarity. Some of my questions, concerns, and suggestions are as follows:

1. Abstract – The opening sentences in the objectives and methods are very similar. Please revise the objectives to specify that the purpose of this qualitative study was to identify enablers and barriers to the uptake of HIV self-testing among sexually active youth in Nigeria using journey map methodology.

2. Introduction – It would be helpful to clarify the meaning of “effective self-testing in the private sector” in the introduction. Does this refer to the distribution of HIV self-testing kits by manufacturers or pharmacies? Would the kits be available for free, or would they have to be purchased? Is this already being done in any states in Nigeria? What types of HIV self-tests are commercially available – oral fluid or finger-stick blood or both?

3. Methods – How were participants recruited? What were the inclusion and exclusion criteria?

4. Methods – Please provide a brief description (2-3 sentences) of the journey map methodology along with appropriate citations for the benefit of readers unfamiliar with this approach.

5. Results – It would be helpful to include a table summarizing the demographic characteristics of the 80 participants. Figure 1 does include their geographic distribution, but no figures have been presented. For example, how many men and women were in the sample? What was the age range? How many lived in urban and peri-urban areas?

6 Discussion – Is there a reason why the results might not be subject to social desirability bias?

Reviewer #2: The authors should streamline their manuscript to focus on Enablers and Barriers to effective HIV self-testing in Private sector. Currently the manuscript is disorganized with irrelevant information,which doesn't address the enablers/barriers to effective HIVST. Also the authors should explicitly describe how was the Journey Map method operationalized in this study so a reader can follow. The author should also highlight what was the role of the private sector in this study. Did the author want to describe the role private health delivery systems? The author should use scientific writing, currently the manuscript is full of irrelevant words which doesn't add any value to the paper. The authors should check for grammar and spellings before their next submission.

6. PLOS authors have the option to publish the peer review history of their article (what does this mean?). If published, this will include your full peer review and any attached files.

Reviewer #1: No

Reviewer #2: No

---

## [Author Response · Author response to Decision Letter 0]

16 Jan 2023

Society for Family Health,

No. 8 Port Harcourt Crescent

Abuja, Nigeria

November 15th 2022

The Editor-in-Chief,

Editorial Office

PLOS ONE

Dear Editor-in-Chief,

We thank the academic editor and the reviewers for the comments to improve on our manuscript [PONE-D-22-23769]. Please see below our itemized point-by-point responses has been highlighted. Moreover, in the revised manuscript, all changes have been tracked to indicate the revised sections. We would be happy to clarify any aspect of our responses if needed. 

Best Regards,

Yusuf H. Wada

Corresponding Author

hwada@sfhnigeria.org

Academic Editor

We thank the academic editor for this comment, we have updated the manuscript with the PLOS ONE’s style requirement as advised. 

“This study was possible by the generous support of Children’s Investment Fund Foundation (CIFF) - Global Fund partnership through the Population Services International (PSI). The contents herein are the sole responsibility of the authors and do not necessarily reflect the views of SFH, PSI or CIFF. The funders had no role in study design, data collection and analysis, decision to publish or preparation of the manuscript. We appreciate all members of the SFH SHIPS team, NACA, and ANAYD for their support during the data collection. We are most grateful to all the participants who took part in the study.”

“This study was possible by the generous support of Children’s Investment Fund Foundation (CIFF) - Global Fund partnership through the Population Services International (PSI). The contents herein are the sole responsibility of the authors and do not necessarily reflect the views of SFH, PSI or CIFF. The funders had no role in study design, data collection and analysis, decision to publish or preparation of the manuscript. “

We thank the academic editor for this. This have been addressed. 

“None”

We thank the academic editor for this. This have been addressed. 

We thank the academic editor for this. 

5. Please ensure that you refer to Figure 1 in your text as, if accepted, production will need this reference to link the reader to the figure.

We thank the academic editor for this. We have changed in figure to a new one as reflected in the fig 1 uploaded. 

6. We note that Figure 1 in your submission contain [map/satellite] images which may be copyrighted. All PLOS content is published under the Creative Commons Attribution License (CC BY 4.0), which means that the manuscript, images, and Supporting Information files will be freely available online, and any third party is permitted to access, download, copy, distribute, and use these materials in any way, even commercially, with proper attribution. For these reasons, we cannot publish previously copyrighted maps or satellite images created using proprietary data, such as Google software (Google Maps, Street View, and Earth). For more information, see our copyright guidelines: http://journals.plos.org/plosone/s/licenses-and-copyright.

 a. You may seek permission from the original copyright holder of Figure(s) [#] to publish the content specifically under the CC BY 4.0 license. 

We thank the academic editor for this. We have provided a replacement map as directed. 

Reviewers' comments:

Reviewer's Responses to Questions

Comments to the Author

1. Is the manuscript technically sound, and do the data support the conclusions?

Reviewer #1: Yes

Reviewer #2: Partly 

2. Has the statistical analysis been performed appropriately and rigorously?

Reviewer #1: N/A

Reviewer #2: Yes 

3. Have the authors made all data underlying the findings in their manuscript fully available?

Reviewer #1: No

Reviewer #2: Yes 

4. Is the manuscript presented in an intelligible fashion and written in standard English?

Reviewer #1: No

Reviewer #2: No

5. Review Comments to the Author

Reviewer #1: 

“Enablers and Barriers to Effective HIV Self-testing in the Private Sector among Sexually active youths in Nigeria: A Qualitative Study using Journey Map methodology” presents qualitative data from in-depth interviews and focus group discussions with 80 participants involved in market research in Lagos, Anambra, and Kano states from January 2021 to October 2021. Although this is an important area of investigation, the current writing is weak and has numerous grammatical errors, making portions of the manuscript difficult to comprehend. I strongly recommend utilizing the services of a copyeditor to enhance clarity. Some of my questions, concerns, and suggestions are as follows:

We thank the reviewer 1 for this comment. The manuscript has been edited by a copyeditor to enhance clarity as advised. 

1. Abstract – The opening sentences in the objectives and methods are very similar. Please revise the objectives to specify that the purpose of this qualitative study was to identify enablers and barriers to the uptake of HIV self-testing among sexually active youth in Nigeria using journey map methodology.

We thank the reviewer 1 for this comment, the objectives of the study have been rephrased as advised.

2. Introduction – It would be helpful to clarify the meaning of “effective self-testing in the private sector” in the introduction. Does this refer to the distribution of HIV self-testing kits by manufacturers or pharmacies? Would the kits be available for free, or would they have to be purchased? Is this already being done in any states in Nigeria? What types of HIV self-tests are commercially available – oral fluid or finger-stick blood or both?

We thank reviewer 1 for this comment. Within the context of this manuscript, effective self-testing in the private sector means the effective utilization of HIVST kits and linkage to sexual reproductive health services and to HIV prevention by pharmacies (who are only approved to provide those services), while the distribution of HIV self-testing at the retail level is by the pharmacies. This is in line with the revised National HIV and AIDS strategic framework 2019-2021 as a priority policy and porgarmmatic approach to HIV response in Nigeria (NACA. Revised National HIV and AIDS Strategic Framework 2019-2021. Abuja: Nigeria. National Agency for Control of AIDS; 2019). We also updated the manuscript to reflect the context of effective self-testing in the private sector. The kits would have to be purchased at the pharmacies and currently being sold in different pharmacies in Lagos, FCT, Rivers, Akwa-Ibom, Anambra and other states in Nigeria. Therefore, this pilot study finding is set to to provide evidencebased date and influence government, policy makers, donors and business/investment case to scale up free, incentivized or subsidized HIVST kits and for successful future HIVST campaigns in the private sector. There are many types of HIVST kits commercially available in Nigeria such as OraQuick (oral-fluid based), Mylan (blood-based), Insti (blood-based), DrGregs (blood based) and 3-H viral blood check (blood-based). There have been some discrepancies from regulatory bodies on which is approved or not, reason we didn’t discuss earlier in our introduction. All comment raised by the reviewer have been inputted in the introduction part of the manuscript (line 77-84). 

3. Methods – How were participants recruited? What were the inclusion and exclusion criteria?

We thank reviewer 1 for this comment. The recruitment of participant was using a clearly spelt-out eligibility criteria (see below) and using a convenience sampling from a list of participants who fulfilled the criteria using a CBOs & CSOs drafted list and were willing to write an informed consent, which was carried out in accordance with ethical guidelines of 1975 declaration of Hesinki Declaration of 1975, revised 2000. 

Inclusion criteria were being either male or female, aged between 18 and 29, being sexually active, currently a resident of urban or peri urban setting in Nigeria, has ever taken an HIVST and other group who has never taken an HIVST, willing and able to provide verbal oral informed consent, and willing to consent to an audio recorded session. 

We have updated the participant selection and recruitment part to reflect the comment being raised (line 125-133). 

4. Methods – Please provide a brief description (2-3 sentences) of the journey map methodology along with appropriate citations for the benefit of readers unfamiliar with this approach.

We thank reviewer 1 for this comment. We have updated the method with a journey map section as recommended. 

5. Results – It would be helpful to include a table summarizing the demographic characteristics of the 80 participants. Figure 1 does include their geographic distribution, but no figures have been presented. For example, how many men and women were in the sample? What was the age range? How many lived in urban and peri-urban areas?

We thank reviewer 1 for this comment. We have updated the result section with (Table 1) to reflect the gender, age range and geographic distribution.

6 Discussion – Is there a reason why the results might not be subject to social desirability bias?

We thank the reviewer for this comment. We have already reported in our limitations that our study might be subject to social desirability bias, but we try as much to reduce it by using indirect questions, self-completion, the use of proxy users (both the users and non-users), and random sampling from list of participants presented. This have also been updated in the manuscript. 

Reviewer #2: The authors should streamline their manuscript to focus on Enablers and Barriers to effective HIV self-testing in Private sector. Currently the manuscript is disorganized with irrelevant information, which doesn't address the enablers/barriers to effective HIVST. 

We thank the reviewer 2 for the comment. We have ie 

Also, the authors should explicitly describe how was the Journey Map method operationalized in this study so a reader can follow. 

We thank reviewer 2 for this comment. This have been reflected in the methodology section by adding a journey map method, how participants were recruited and the inclusion and exclusion criteria. 

The author should also highlight what was the role of the private sector in this study. Did the author want to describe the role private health delivery systems? 

We thank the reviewer 2 for this comment. We have updated the introduction section to reflect the context of private sector in this study. 

The author should use scientific writing, currently the manuscript is full of irrelevant words which doesn't add any value to the paper. The authors should check for grammar and spellings before their next submission.

 We thank the reviewer for this comment. We have employed the service of a copy writer and now reflect in the updated manuscript. 

6. PLOS authors have the option to publish the peer review history of their article (what does this mean?). If published, this will include your full peer review and any attached files. Do you want your identity to be public for this peer review? For information about this choice, including consent withdrawal, please see our Privacy Policy.

Reviewer #1: No

Reviewer #2: No

---

## [Decision Letter · Decision Letter 1]

29 Mar 2023

PONE-D-22-23769R1Enablers and Barriers to Effective HIV Self-testing in the Private Sector among Sexually active youths in Nigeria: A Qualitative Study using Journey Map methodologyPLOS ONE

Dear Dr. Hassan Wada,

Thank you for submitting your manuscript to PLOS ONE. After careful consideration, we feel that it has merit but does not fully meet PLOS ONE’s publication criteria as it currently stands. Therefore, we invite you to submit a revised version of the manuscript that addresses the points raised during the review process.

We look forward to receiving your revised manuscript.

Kind regards,

Adetayo Olorunlana, Ph.D.

Academic Editor

PLOS ONE

Journal Requirements:

Reviewers' comments:

Reviewer's Responses to Questions

**Comments to the Author**

1. If the authors have adequately addressed your comments raised in a previous round of review and you feel that this manuscript is now acceptable for publication, you may indicate that here to bypass the “Comments to the Author” section, enter your conflict of interest statement in the “Confidential to Editor” section, and submit your "Accept" recommendation.

Reviewer #2: All comments have been addressed

Reviewer #3: All comments have been addressed

2. Is the manuscript technically sound, and do the data support the conclusions?

Reviewer #2: Yes

Reviewer #3: Yes

3. Has the statistical analysis been performed appropriately and rigorously? 

Reviewer #2: Yes

Reviewer #3: Yes

4. Have the authors made all data underlying the findings in their manuscript fully available?

Reviewer #2: Yes

Reviewer #3: Yes

5. Is the manuscript presented in an intelligible fashion and written in standard English?

Reviewer #2: Yes

Reviewer #3: Yes

6. Review Comments to the Author

Reviewer #2: The authors should streamline their manuscript to focus on Enablers and Barriers to effective HIV self-testing in Private sector. Currently the manuscript is disorganized with irrelevant information,which doesn't address the enablers/barriers to effective HIVST. Also the authors should explicitly describe how was the Journey Map method operationalized in this study so a reader can follow. The author should also highlight what was the role of the private sector in this study. Did the author want to describe the role private health delivery systems? The author should use scientific writing, currently the manuscript is full of irrelevant words which doesn't add any value to the paper. The authors should check for grammar and spellings before their next submission.

Reviewer #3: In their present manuscript, the authors have documented factors likely to influence young people's uptake of HIV self-testing in Nigeria through a qualitative approach. The objective is clear and a timely question in the context of HIV prevention in West Africa. The method is appropriate, and the authors have correctly addressed previous reviewer suggestions. Here are a few comments to consider to further strengthen the manuscript:

•The authors should ensure that the paper is formatted according to the COREQ checklist.

Methods:

•Consider moving the “Recruitment and participants selection” section after the “Journey map operation” section.

•Line 121: Describe the HIV prevention stage. Refer to the following articles that describes HIV prevention continuum. A. McNairy, Margaret L, and Wafaa M El-Sadr. “A paradigm shift: focus on the HIV prevention continuum.” Clinical infectious diseases: an official publication of the Infectious Diseases Society of America vol. 59 Suppl 1,Suppl 1 (2014): S12-5. doi:10.1093/cid/ciu251. B. Horn, Tim et al. “Towards an integrated primary and secondary HIV prevention continuum for the United States: a cyclical process model.” Journal of the International AIDS Society vol. 19,1 21263. 17 Nov. 2016, doi:10.7448/IAS.19.1.21263.

•Where was the interview conducted and by whom?

•Include more description about the study setting. Anambra, Kano and Lagos states. Perhaps mention that these states were identified as PEPFAR priority states given evidence of high HIV burden and unmet needs for HIV/AIDS treatment services

•How was data triangulated?

Result

•Consider deleting “The journey map is the process with each stage having its own characteristics using the person’s experience to identify problems and suggest an improvement. However, the person perceives this as a real journey that can be used to improve each stage of HIV self-testing.” This has already bee described in the methods section.

•Line 213 revise to “While the barriers identified at the attracting stage..”

•Line 250, include the dollar equivalence of the naira

7. PLOS authors have the option to publish the peer review history of their article (what does this mean?). If published, this will include your full peer review and any attached files.

Reviewer #2: No

Reviewer #3: **Yes: **Chisom Obiezu-umeh

---

## [Author Response · Author response to Decision Letter 1]

3 Apr 2023

Society for Family Health,

No. 8 Port Harcourt Crescent

Abuja, Nigeria

March 30th 2023

The Editor-in-Chief,

Editorial Office

PLOS ONE

Dear Editor-in-Chief,

We thank the academic editor and the reviewers for the comments to improve on our manuscript [PONE-D-22-23769]. Please see below our itemized point-by-point responses has been highlighted. Moreover, in the revised manuscript, all changes have been tracked to indicate the revised sections. We would be happy to clarify any aspect of our responses if needed. 

Best Regards,

Yusuf H. Wada

Corresponding Author

hwada@sfhnigeria.org

Reviewer #2: 

The authors should streamline their manuscript to focus on Enablers and Barriers to effective HIV self-testing in Private sector. Currently the manuscript is disorganized with irrelevant information, which doesn't address the enablers/barriers to effective HIVST. Also, the authors should explicitly describe how was the Journey Map method operationalized in this study so a reader can follow. The author should also highlight what was the role of the private sector in this study. Did the author want to describe the role private health delivery systems? The author should use scientific writing, currently the manuscript is full of irrelevant words which doesn't add any value to the paper. The authors should check for grammar and spellings before their next submission.

We thank the reviewer for this comment. We have reviewed how the Journey map was operationalized (line 112-125), describe the role of private sector in the manuscript, reviewed by a native speaker for irrelevant word and updated the format/logical flow using the COREQ checklist which was adopted from Tong A, Sainsbury P, Craig J. Consolidated criteria for reporting qualitative research (COREQ): a 32-item checklist for interviews and focus groups. International Journal for Quality in Health Care. 2007. Volume 19, Number 6: pp. 349 – 357 as highlighted by the other reviewer. 

Reviewer #3:

In their present manuscript, the authors have documented factors likely to influence young people's uptake of HIV self-testing in Nigeria through a qualitative approach. The objective is clear and a timely question in the context of HIV prevention in West Africa. The method is appropriate, and the authors have correctly addressed previous reviewer suggestions. Here are a few comments to consider to further strengthen the manuscript:

•The authors should ensure that the paper is formatted according to the COREQ checklist.

We thank the reviewer for this comment. We have formatted the manuscript according to the COREQ checklist and updated as appropriate. The COREQ Checklist was adopted from Tong A, Sainsbury P, Craig J. Consolidated criteria for reporting qualitative research (COREQ): a 32-item checklist for interviews and focus groups. International Journal for Quality in Health Care. 2007. Volume 19, Number 6: pp. 349 – 357. 

Methods:

•Consider moving the “Recruitment and participants selection” section after the “Journey map operation” section.

We thank the reviewer for this comment. We have revised as advised with the recruitment and participants selection after the journey map operation section. 

•Line 121: Describe the HIV prevention stage. Refer to the following articles that describes HIV prevention continuum. A. McNairy, Margaret L, and Wafaa M El-Sadr. “A paradigm shift: focus on the HIV prevention continuum.” Clinical infectious diseases: an official publication of the Infectious Diseases Society of America vol. 59 Suppl 1,Suppl 1 (2014): S12-5. doi:10.1093/cid/ciu251. B. Horn, Tim et al. “Towards an integrated primary and secondary HIV prevention continuum for the United States: a cyclical process model.” Journal of the International AIDS Society vol. 19,1 21263. 17 Nov. 2016, doi:10.7448/IAS.19.1.21263.

We thank the reviewer for this comment. We have referred to the article and revised as appropriate (see line 126-131 and reference 18-20). 

•Where was the interview conducted and by whom?

We thank the reviewer for this comment. We already have that captured in the manuscript (line 157- 162). We have also included a sentence and updated the manuscript on who conducted the interview and by whom. 

“The project team from SFH and Busara who were part of the face validity team carried out the qualitative research and collected data in the four dominant language of Nigeria (English, Igbo, Hausa, and Yoruba) using the IDIs and FGDs guides. IDIs were done in English over the phone, while FGDs were conducted in-person for the sexually active youths using the four languages in the three states. We conducted gender-specific FGDs - one FGD was either all men or all women, but we mixed HIVST users and HIVST non-users. During both the IDIs and FGDs, we asked respondents open-ended questions regarding their opinions and perceptions of HIVST, and for IDIs with HIVST users, we asked questions about their past experiences with taking up and using HIVST kits”. 

•Include more description about the study setting. Anambra, Kano and Lagos states. Perhaps mention that these states were identified as PEPFAR priority states given evidence of high HIV burden and unmet needs for HIV/AIDS treatment services

We thank the reviewer for this comment. We have included the brief description of the study setting and identify them as PEPFAR priority states and other supporting information as advised (see line 110-117). 

•How was data triangulated?

We thank the reviewer for this comment. We have included how the data was triangulated in our manuscript as advised (see line 167-170). 

Result

•Consider deleting “The journey map is the process with each stage having its own characteristics using the person’s experience to identify problems and suggest an improvement. However, the person perceives this as a real journey that can be used to improve each stage of HIV self-testing.” This has already been described in the methods section.

We thank the reviewer for this comment. This have been revised and deleted this section. 

•Line 213 revise to “While the barriers identified at the attracting stage.”

We thank the reviewer for this comment. We have revised as advised. 

•Line 250, include the dollar equivalence of the naira 

We thank the reviewer for this comment. We have included the dollar equivalence of the naira (US$ 0.2-2).

---

## [Editor Report · Decision Letter 2]

13 Apr 2023

Enablers and Barriers to Effective HIV Self-testing in the Private Sector among Sexually active youths in Nigeria: A Qualitative Study using Journey Map methodology

PONE-D-22-23769R2

Dear Dr. Hassan Wada,

We’re pleased to inform you that your manuscript has been judged scientifically suitable for publication and will be formally accepted for publication once it meets all outstanding technical requirements.

Kind regards,

Adetayo Olorunlana, Ph.D.

Academic Editor

PLOS ONE
---

## [Editor Report · Acceptance letter]

17 Apr 2023

PONE-D-22-23769R2 

Enablers and Barriers to Effective HIV Self-testing in the Private Sector among Sexually active youths in Nigeria: A Qualitative Study using Journey Map methodology 

Dear Dr. Hassan Wada:

I'm pleased to inform you that your manuscript has been deemed suitable for publication in PLOS ONE. Congratulations! Your manuscript is now with our production department. 

Kind regards, 

on behalf of

Associate Professor Adetayo Olorunlana 

Academic Editor

PLOS ONE